# Making Rural Micro-Regeneration Strategies Based on Resident Perceptions and Preferences for Traditional Village Conservation and Development: The Case of Huangshan Village, China

**Xuesong Xi [1], Haiyun Xu [2,*], Qiang Zhao [1] and Guohan Zhao [3]**

[1] Department of Agricultural Structure and Bioenvironmental Engineering,
College of Water Resources and Civil Engineering, China Agricultural University, Beijing 100083, China;
xixuesong@cau.edu.cn (X.X.); Zhaoqiang@cau.edu.cn (Q.Z.)

[2] Department of Landscape Architecture, College of Architecture and Urban Planning,
Beijing University of Civil Engineering and Architecture, Beijing 102627, China

[3] Department of the Built Environment, Aalborg University, Thomas Manns Vej 23,
9220 Aalborg Øst, Denmark; guohanz@build.aau.dk

* Correspondence: xuhaiyun@bucea.edu.cn; Tel.: +(86)-18511709902

**Abstract:** Micro-regeneration is a gradual renewal strategy that uses small-scale interventions to improve the quality of the living environment and local community, as well as spur industrial development. It is the small-scale interventions that have continued to make micro-regeneration a viable economic rural renewal approach for traditional village conservation and development. As such, in this work we explore potential micro-regeneration strategies and promotions based on assessments of public perception and preferences in an "unlisted" traditional village in China (i.e., an area with limited investment for conservation compared to "listed", renowned traditional villages). We aim to identify the most perceptible modes of village transformation and industrial development for rural micro-regeneration strategies in the Huangshan traditional village of China. We also tested how the social character of respondents significantly affected their preferences in this regard. The public participatory mapping results illustrated a spatially clustered pattern made up of small spaces and individual buildings demanding micro-regeneration interventions. The survey based on 150 residents living around these sites disclosed that a unified repair approach subsidized by government and traffic condition improvements are the most recognized modes of village transformation, and the tourism is the most perceived and preferred method for industrial development. Significant differences between public perceptions and preferences of both village transformation and industrial development were identified corresponding to gender and income demographics, while village transformation perceptions change is dependent on age. Therefore, our study demonstrates evidence-based recommendations for active and effective rural micro-regeneration practices.

**Keywords:** traditional villages; micro-regeneration; urban acupuncture; public perceptions and preferences; cultural landscape management; rural development; conservation planning

## 1. Introduction

At the start of the new millennium, China's central government began to develop a series of laws and policies that would target the conservation and revitalization of rural traditional villages all over the country [1]. Many of these traditional villages, having been built before 1949, continue to benefit from a wide range of historical, cultural, and architectural advancements, such as architectural style and location; yet, many have progressed without significant alterations to the social/community environment. To be sure, many of these villages are still home to unique folk customs which tell stories of the

local cultural landscape spanning a century or more. Each still serves their people, with the community coming together as a type of 'micro-civilization' imbued with the value of inheritance and the meanings of culture to form an integral part of China's rural cultural landscape [1–3].

Over the last several decades, however, even prior to the central government taking charge of conservation efforts around the country, China's rural cultural landscape has suffered somewhat under the weight of wide-scale urbanization and globalization. While a community still exists, the more traditional features of these villages and their rural cultural landscapes are threatened due to a variety of factors, including farmland abandonment and population loss due to increased industrialization.

Thankfully, this issue has since attracted increased political attention due to committed decision-makers who focus their efforts on the promotion of rural renewal strategies that support the transformation and revitalization of these traditional villages. In 2008 for instance, the Regulations for the Protections of Historical Cities and Villages was passed, which lawfully names Chinese traditional villages as deeply important cultural heritage sites that require ongoing conservation efforts during rural planning and development phases. At present, there are 211 traditional villages listed by the Chinese central government as being "National Famous Historical and Cultural Villages", that are rich in cultural, historical, and natural resources. These traditional villages have been named as models for the protection, conservation, and revitalization of China's traditional rural cultural landscape, and have since received a wide range of public investment and financial support from both central and local governments.

Given the focus that many of these traditional villages now have on conservation and revitalization, it may then be of interest to note that there are more than 630,000 traditional villages in China which have not been named by the government as specifically requiring conservation. In other words, there are limited investments and resources available to support more than 97% of the traditional villages which exist in China. And while in 2018 the Chinese government did succeed in their promotion of a national plan for rural renewal—specifically to support a rural business, improve quality of living, and facilitate community—that plan does not address the majority of rural traditional villages in China, though it does admittedly build a variety of rural-based strategies into its frameworks, such as urban-rural harmonious development and environmental improvements. Nevertheless, the reality is that without a more comprehensive focus on all traditional villages, China's rural traditional village conservation efforts will continue to be plagued by limited local financial budgets, which themselves demand a suitable economic strategy toward the protection and revitalization of these cultural heritage sites.

In this context, it becomes necessary to facilitate an exploration of suitable rural renewal approaches to improve not just space functions, but also resident well-being for these 'unlisted' traditional villages. Such an approach first demands consultation with residents about their needs, perceptions, and preferences as they relate to local planning, and second requires an activation protocol to stimulate industrial development, community relations, and economic vitality to support the village's overall rural renewal. A substantial amount of research has addressed the understanding that the public view of the local environment could support improved rural renewal and development [4,5]. In addition, existing studies have addressed the notion that rural industrial development directly promotes the growth of rural communities related to these industries [6]. To best stimulate rural industries and economic vitality, the government should organize local isolated household industries into clusters, empower the local community, provide training courses, effectively sustain local skill sets, and encourage local innovations through sustained, collaborative relationships with residents [6,7]. These relationships will also strengthen community relations and cohesion to achieve rural renewal and prosperity [8]. However, very little literature and practical cases are known concerning the engagement of public perceptions about community building, or industrial and economic stimulation via rural renewal practices in China. A comprehensive analysis of rural renewal studies in this

region has found that most are focused on top-down governance and policies, whereas more suitable and practical approaches to engaging public views—to allow improved contributions to local communities and to improve industrial development—have remained outside the scope of most rural renewal considerations in China, especially for unlisted villages with limited land-development budgets.

We thus performed our practice in one such 'unlisted' traditional village to engage resident perceptions and preferences on rural renewal projects that incorporate micro-regeneration, which is itself a targeted, gradual, regional revitalization approach to urban planning via small-scale interventions. Our study conferred with local respondents about their demands for village transformation and industry improvement interventions. We developed a case study in Huangshan village, an 'unlisted' cultural center with a burgeoning tourism sector in Yiwu City. The village was in great need of a viable micro-regeneration strategy to stimulate the local community and the industry for traditional village conservation and cultural landscape development. Our study aims to identify how resident social backgrounds affect their perceptions and preferences regarding micro-regeneration in their local area. We hypothesize that residents' gender, age, and income could make significant differences towards their perceptions and preferences concerning village transformations and industrial development regarding the micro-regeneration in traditional village conservation.

Our study raises the following research questions:

(1) Where do residents spatially localize the sites demanding micro-regeneration they perceive in the whole village?
(2) Are the public perceptions and preferences on village transformations affected by their gender, age and income?
(3) Are the public perceptions and preferences on industrial development affected by their gender, age and income?

Given our intention to improve on existing rural cultural landscape conservation efforts, and to provide inspiration for those 'unlisted' traditional villages, this study intends to create a convincing case for participant-focused micro-revitalization in the face of barriers to cultural landscape revitalization. We argue in this paper for the improved development of potential rural micro-regeneration strategies, and for the contribution of unlisted traditional village conservation in the context of China's current rural renewal practices and cultural history.

## 2. Rural Development Based on Micro-Regeneration Strategy

A wide range of rural renewal strategies has been used by land-use planners and developers to guide rural development, which significantly improves rural development and local economic policies worldwide. From neo-endogenous rural development, to community planning, to micro-regeneration, such strategies continue to provide theoretical support for current renewal policy-making from a social standpoint. Indeed, reliable regional planning practices, such as community planning and neo-endogenous development, have served many development initiatives well across the world. For instance, neo-endogenous rural development has found success in many European rural revitalization studies [9–12] largely due to an approach that altogether improves access via the participation of local actors, relies on external networks to benefit local resource allocation, and even balances competition against contributions to stimulate local industry [13]. Community planning also requires major strides to ensure a more livable environment via land-use regulations, comprehensive design guidelines, and well-developed finance packages [14,15].

Still, while many such strategies continue to focus on grand-scale revitalization strategies, it is only micro-regeneration that takes a gradual, resident-focused approach to urban or rural renewal. Such micro-scale strategies consider current gaps—from a physical and social standpoint—to discover more suitable space change applications for differing

scales and functions, especially in locations where the developmental direction is still very much required.

It is for these reasons and more that micro-regeneration as a strategy for rural-urban renewal has been gaining scholarly attention in the present day. Defined as a targeted, gradual revitalization approach that relies on small-scale interventions to improve the quality of a given living environment, micro-regeneration places a primary emphasis on improvements to the local living environment, resident welfare, community governance regarding local regeneration, and industrial development. When compared to other renewal strategies like community planning and neo-endogenous rural development, micro-regeneration provides a more specific work focus on the dimension of physical space—in effect, on small-scale projects over abstract policy. And while community planning focuses on the necessity of the participation of local community members to achieve community control and autonomy, government affiliates and planners could not promote their community planning strategies without involvement from local groups. For micro-regeneration strategies, community planning is considered integral to the success of the approach itself. Yet, existing practices have shown that micro-regeneration most likely were promoted via local governments and planners, or via stakeholders from outside the given area, such as external capital holders, or high-level authority figures and planning consultants [16–18], whereas the cooperation with local residents brings better results. Thus, micro-regeneration strategies which aim to create small-scale physical space interventions that require a flexible stakeholder-ship will provide the most targeted, practical, efficient, and flexible regional renewal strategies. These strategic approaches will also find added benefit from lower costs associated with small-space transformations, adding to their viability as stand-alone economic renewal strategies. As early as 2012, the Ministry of Housing and Urban Development began promoting micro-regeneration as a mainstay of economic and infrastructural renewal, and in response, many regional administrators have gradually abandoned their more large-scale and rapid urban transformation practices in favor of medium-sized and gradual renewal strategies that target local regeneration [19,20].

As an added point of interest, the Chinese government's focus on micro-regeneration is largely based on an existing theory called 'urban acupuncture' [21], a social-environmental theory that was inspired by traditional Chinese acupuncture. Here, the framework is a process that uses small-scale physical space interventions to effectively transform larger community and regional development initiatives. In 1980, these small-scale physical space interventions contributed to the intense renovation of Barcelona's urban landscape, and ultimately, the revitalization of its local urban identity [22,23]. Since then, a great number of scholars and regional decision-makers have highlighted the benefits and contributions inherent to using urban acupuncture toward regional development practices around the globe, with land developers using this knowledge to transform communities of all shapes and sizes. For instance, in Mexico, such practices have a high potential to help residents alleviate stress, create improved social pathologies, as well as promote the political disengagement of community settlements [24]. What is more, these small-scale physical space interventions have reinvigorated social activity in the region, stimulating the historical and cultural rehabilitation of San Paulo [25].

In sum, the main emphasis of urban acupuncture is on the limited expansion of micro-scale physical space, yet insufficient attention has been paid to the promotion of community governance on a social and cultural scale. In contrast, micro-regeneration-as-renewal strategies aim to improve community governance via the transformation of specific physical space interventions, and the deployment of social interventions for the management of community cohesion and cooperation enhancements. Such approaches also facilitate improvements to local industrial development, which is partly why 'urban acupuncture' theory has inspired micro-regeneration theory via its focus on small, existing spaces in already active communities. To be sure, micro-regeneration has and continues to broaden the cultural and social dimensions of many environments, all of which are

required to successfully implement the practices of 'urban acupuncture in general, and micro-regeneration specifically.

Perhaps inspired by the global adoption of early urban acupuncture practices, administrators and researchers from China's urban planning and community development branches have since promoted micro-regeneration strategies within a local planning context [26]. During this process, scholars began to record a new understanding of micro-regeneration strategies as being so much more than simply the development of small-scale physical space interventions; they also promote inclusive community planning, which includes the protection of local history, the mobilization of public participation, and the identification of a 'sense of place' [17,27,28]. These practices highlight the nature of micro-regeneration in the protection of local history, and the progressive promotion of large-scale community-level changes due to a singular catalyst: urban economic investment [28]. Ever since the first micro-regeneration project in Beijing succeeded in its historical block renewal, most current micro-regeneration practices in China are regarded as effective economic strategies that should be implemented during urban governance and urban community development projects across the country [17,29,30].

Naturally, and to the point of this paper, many important areas are rarely considered for rural development and even micro-regeneration in China. When compared to urban communities, rural communities have a closer social order and networking relationship to local land-use rights and a sense of place, which directly relates to these residents' daily way of life. The design and development of small-scale physical space interventions via micro-regeneration strategies in rural communities thus requires greater consideration of the values that these residents hold. In other words, micro-regeneration strategies must consider resident values, as they are key stakeholders who will ultimately determine the success of a given project beyond government decision-makers and professionals as the common stakeholders involved in rural renewal projects in China. The continued exploration and development of rural micro-regeneration strategies must therefore be based on resident perceptions and perspectives, and specifically on the integration and activation of local natural and cultural resources, or the introduction of physical, industrial, and spatial elements that further promote rural development. At present, however, China's decision-makers dominate a mainstream, top-down spatial planning process with little representation from residents [31–33]. While this autonomy has resulted in residents being invited to a small number of public hearings and consultations [34,35], most spatial planning strategies proceed without much knowledge of resident perceptions and preferences—an outcome that is far from the primary aims of micro-regeneration strategies as a whole.

When we look back to the pre-determinants for micro-regeneration philosophy as it relates to urban and rural planning, the unifying theme produced through such initiatives has been a long-term, gradual approach that focuses on a physically small space as an object with high transformative potential for both resident well-being and economic investment. Again, micro-regeneration intends to link the physical improvement of a given space and the social wellbeing of the people who live there, all in the pursuit of rural renewal and transformative development. Such approaches aim to produce multi-function interventions that sustain community needs and even increase community cohesion. In light of this resident-focused, micro-environmental approach, there is much to be learned from any study which produces a measure for the qualitative and quantitative aspects of the local community and infrastructural improvements—an aim this study shares.

## 3. Methods and Materials

### 3.1. Case Study Description

Huangshan traditional village is 11.5 km from Yiwu city, which is a small commodity trade center in the Yangtze River Delta region of Zhengjiang Province, China (Figure 1). The village is situated in the valley of Bei Mountain, which runs west to east on the southern side of Phoenix Stream. Its territory and location were originally guided by traditional

Chinese Feng Shui theory, which dictates development in tune with "surrounding hills and water".

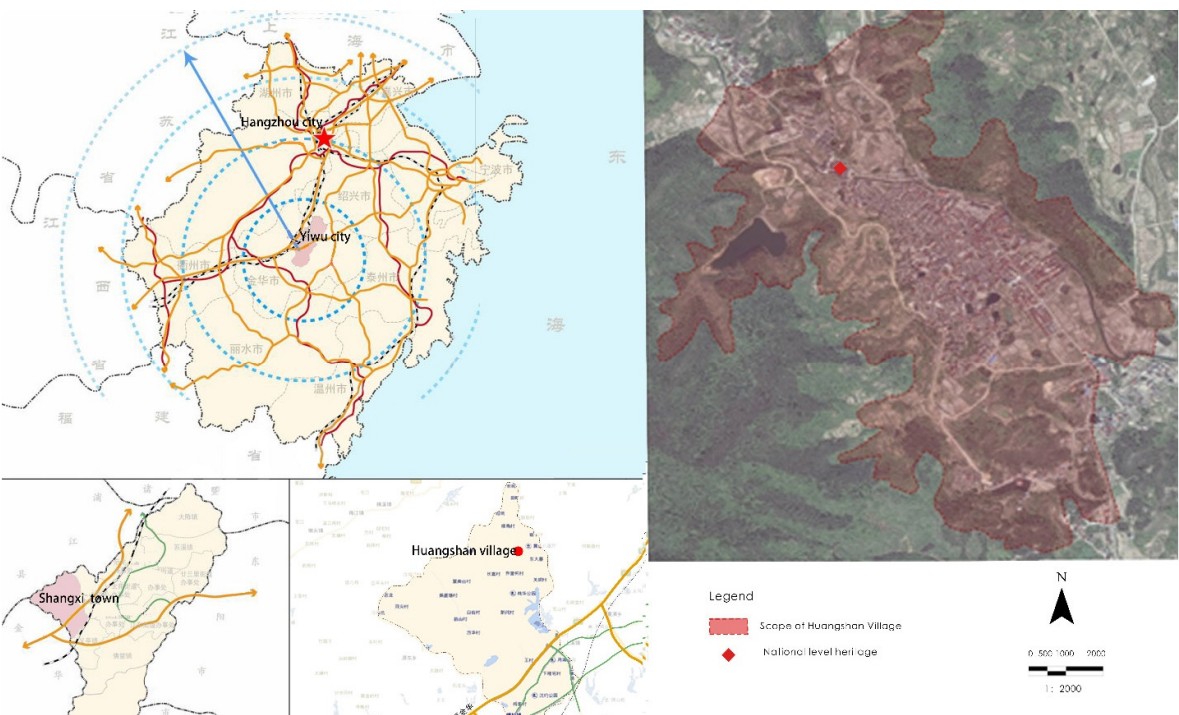

**Figure 1.** The location and territory of Huangshan traditional village.

Huangshan has 1381 households consisting of 3199 people, with the majority of residents bearing the family name Chen. Local landscape features include bodies of water, woodlands, and cultivated land (i.e., orchards and/or vegetable fields). Thus, the main income resources of residents come from the cultivation of fruits and vegetables. There is also active food and handmade goods commerce (e.g., yellow rice wine production) in the area. Moreover, its proximity in the Zhengjiang Province, the home of major corporation Alibaba and one of the most developed E-commerce regions in China, has caused a trickle-down effect, whereby Huangshan village also began to pay attention to local agricultural commerce and production. The village's recorded history dates to the 13th century, from which time the residents of Huangshan village have maintained a culture of appreciation and education to preserve their genealogies.

With the passage of each dynasty, many inhabitants of Huangshan village have also achieved some level of fame, and their deeds further embellish resident genealogies. Thus, this village has a historical record of investment and high status, where the recognition of educational and cultural achievements has taken place. The village buildings and the spatial patterns of both fields and yards are consistent with the ancient Chinese philosophical theories of Feng Shui and Yi Yang, which denote the use of mounts for ancient pools. The wood, stone, and brick carvings on local historical buildings are in the typical folk fashion of the south Yangtze region. These carvings are rich in ancient inscriptions, family instructions, and details. They are not only a fossilized record of the social status of house-holders preserved for posterity but also a record of spiritual and cultural wealth. In 2001 for instance, the ancient hall representing House Chen (called The 'Eightside' Hall) was listed as a protected cultural heritage site worthy of urban regeneration. This identification took place at a time when the Chinese central government had started to recognize certain villages as having historical and cultural importance. After that, Huangshan village began to promote its tourism industry and became an increasingly popular weekend destination for rural tourists in the Yangtze River Region.

However, because Huangshan as a whole was not included on the list of "National Famous Historical and Cultural Villages", there were few conservation strategies that addressed alternative sites aside from The 'Eightside' Hall (Figure 2). As an "unlisted" traditional village, its spatial pattern is following Feng Shui culture and local cultural landscape, and traditional buildings have all been interspersed with massive emerging constructions in response to fast-increasing tourist and resident demands. During this process, traditional building clusters have shifted to accommodate the disorderly expansion of newly-built modern architectures, which are in a state of continuous encroachment. To deal with this challenge, and to address this position, the village committee assessed resident perceptions and preferences with the cooperation of planning consultants from China Agricultural University. Specifically, the assessment addressed resident views on potential micro-regeneration strategies that would protect the local rural landscape, offset the improvement of public well-being, and promote and develop a sustainable tourism industry and rural renewal, all within the bounds of limited budgetary and economic means.

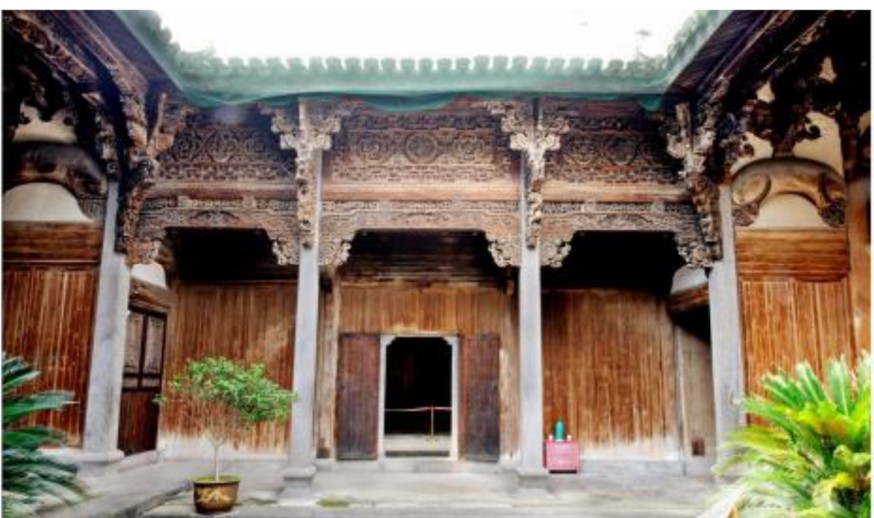

**Figure 2.** National level heritage "The Eightside Hall" in Huangshan Village.

*3.2. Methodology and Data Collection*

We performed our study and combined both quantitative and qualitative methods. Based on document analyses of the literature on local history, we summarized data on the conservation level and location of local important heritage sites (Figure 3). Our research team conducted surveys and qualitative field research in Huangshan village from April to May 2018. During this process, we co-operated with planning consultants from China Agricultural University, who were hired by local government officials to design a plan that would promote traditional village conservation in Huangshan through micro-regeneration. According to the main aims of micro-regeneration as mentioned in part two, our survey and field research consisted of two parts: (1) a public participation map of small scale interventions for micro-regeneration; and (2) questionnaires and semi-structured surveys which focus on resident perceptions and preferences concerning specific micro-regeneration strategies.

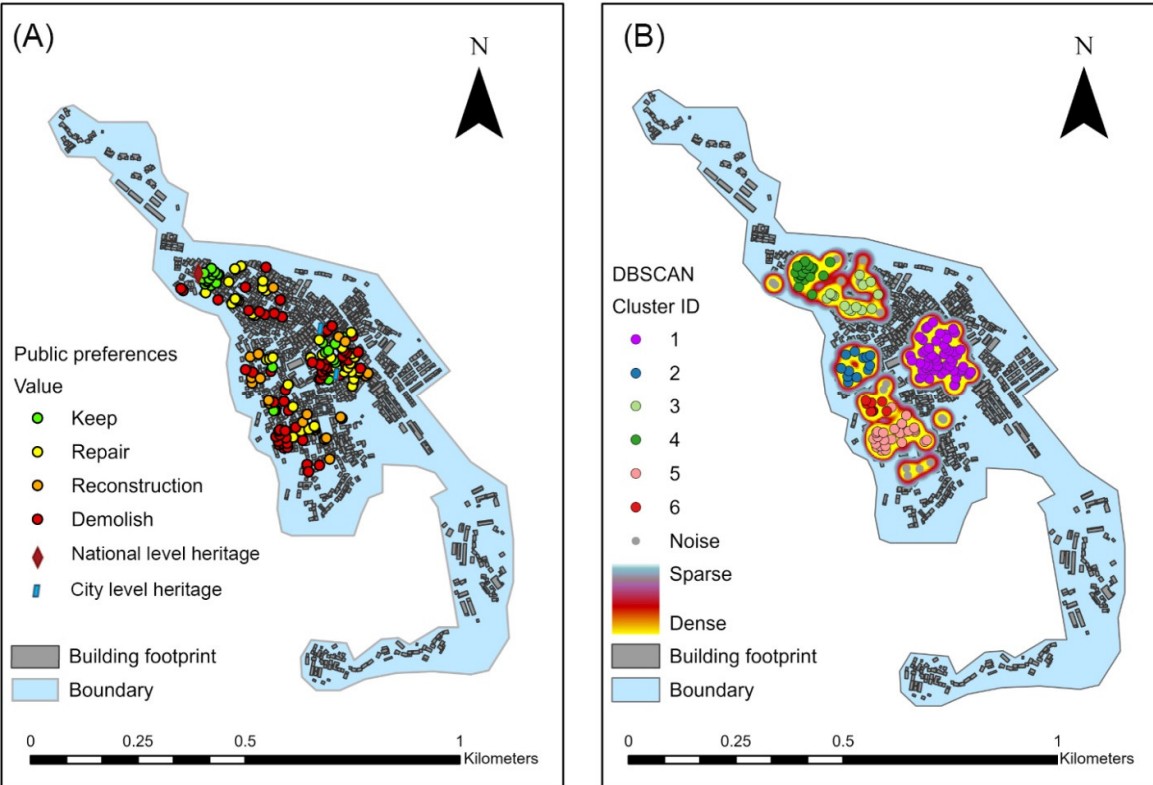

**Figure 3.** The spatial pattern of small spaces and individual buildings per micro-regeneration intervention type (**A**), and the clusters of all mapped point features based on their area densities (**B**).

Firstly, we adopted a snowball sampling method, whereby we invited 59 residents who had settled in Huangshan village, as well as two leaders from the village committee, to our interviews for the public participation mapping survey. According to document analyses of prior studies on micro-regeneration, we summarized current, common micro-regeneration interventions for physical spaces into four types: (1) Keep: pertains to important heritage and historical buildings on local and national conservation lists. This design 'keeps' to the original style of the building, with the maintenance and management of cultural relics required by protection law to improve the surrounding environment; (2) Repair: includes traditional buildings and public spaces that suffer from a current lack of maintenance or upkeep repair. Buildings are improved with raw materials via traditional techniques to repair the outward appearance of these buildings, and to strengthen the multi-functional nature of the buildings as well; (3) Reconstruct: involves the reconstruction of spaces surrounding more modern buildings, which may not be in harmony with the traditional village aesthetic overall. This practice therefore requires renovations to building facades to promote the unity of the traditional village design, as well as a reconstruction of the buildings' functionality and basic facilities to improve the overall environment; and (4) Demolish: includes the demolition of too-modern buildings that seriously affect the overall landscape of the traditional village. Such buildings are demolished and transformed into public spaces and infrastructure developments like green spaces, paths, and shared bike stations. Based on these four micro-regeneration interventions, the planning consultants from China Agricultural University created four related scenarios for the use of these small spaces post-micro-regeneration intervention (Figure A1).

In essence, these scenarios were used to explain the particular ways in which these village transformations were perceived by residents who live in the surrounding area. We then began our interviews by showing respondent-participants these scenarios via visual aids (posters and a paper map) that showed the locations of their familiar landmarks (i.e., the 'Eightside' Hall) and traditional buildings to assist in their identification with and

recognition of the spatial pattern of their village. They were then asked to point out the sites which they believed demand specific, suitable interventions based on micro-regeneration strategies, as well as their perspectives on these (e.g., preference for repairing outward building appearance, or functional reconstructions). We first digitalized the spatial data of public perceptions and preferences into point features in the Geographical Information System (GIS). We then implemented the Density-based Spatial Clustering of Applications with Noise (DBSCAN) method to cluster all the mapped point features into six groups per their area densities. As a result, point features within the "acceptable" buffer were considered in the same group, while those far from other groups were assigned as "noise". Furthermore, the point density of all the point features was visualized as a heat map [36].

According to the contributions of micro-regeneration to physical environment improvements, the management of mode design, and the promotion of industry development, our study thus focused on explorations of how people perceived (1) village transformation (including physical improvements and management methods), and (2) industry development, specifically when micro-regeneration was adopted in a rural context. During the second part of our field research, we interviewed two cadres from the village committee as local decision-makers and were informed by four kinds of current industry and development pathways: (1) Cultivation, (2) Hand-made workshops, (3) Tourism, and (4) E-commerce for local agricultural products.

Next, we invited adult residents who lived around the future micro-regeneration intervention spaces for face-to-face interviews. To ensure as much representativeness as possible, we adopted a cluster random sample method. The entire population was first subdivided into six clusters based on the clustering distribution of four micro-regeneration interventions in the village.

For each cluster, we selected individuals via the adoption of convenience and quota sampling. The respondents were invited to participate in a person-to-person survey about local village transformation and industrial development.

We received 150 valid observations from 201 surveys. Questions in our survey included the basic demographic and socio-economic information of respondents. For local industry, respondents were asked to make suggestions about local industrial development, as well as to record their perceptions and preferences on four kinds of local industry (these based on four industrial directions). Questions included, "How do you feel about the current cultivation/hand workshop/tourism/E-commerce industry?" and "Would you prefer to develop a cultivation/hand workshop/tourism/e-commerce industry in the future?". Moreover, residents were asked to present their perceptions and preferences of village transformation at the intersection of physical improvements and management methods, with questions like "What kind of public infrastructure should be improved or added when new reconstruction begins on these sites?" and "What kind of managing models/methods do you prefer for the repair and restoration of old buildings and public spaces near your residence?" During resident interviews and semi-structured surveys regarding local industrial development, detailed notes on observations, experiences, and resident answers were recorded. We performed descriptive statistical data analyses with STATA 13, and used descriptive analyses to show how socio-demographic characteristics affected resident perceptions and preferences on village transformation and industrial development. These were performed via a Chi-square test to explore whether significant differences exist between respondents concerning the social background and whether those differences affect their perceptions and preferences on village transformation and industrial development. Previous literature addressed various social character variables which affect people's relationships with their living environment—such as green spaces and residential communities [37,38]. Among these, existing studies have indicated that age, gender, and economic status make a significant difference when it comes to public residential satisfaction. It is in this context that we chose to test resident perceptions and preferences on village transformations and industrial development, specifically via four separate micro-regeneration strategies, and on the axes of age, gender, and income.

## 4. Results

### *4.1. Spatial Patterns for Micro-Regeneration Interventions Based on Public Perspectives*

We produced a spatial pattern based on key information relayed about small spaces and individual buildings which demand one or more (with a total of four) micro-regeneration interventions based on public participatory mapping completed by long-settled residents (Figure 3A).

On average, each long-term settled resident mapped three points, for a total of 147 potential target sites for micro-regeneration interventions. The small spaces and individual buildings which demanded demolition (n = 57) were by far the most frequently mapped points, followed by repair (n = 48), keep (n = 26), and restoration (n = 16). Kernel density indicated a clustered distribution of the sites that demand micro-regeneration interventions. We indicted a high spatial intensity for sites where residents preferred to be kept located on the northern side of the village, in areas around the national heritage site, the 'Eightside' Hall. Listed sites for repair, which were the points with the most density, were mapped mostly in areas surrounding two city-level heritage locations. Respondents mentioned the national heritage buildings in well-preserved statements; at the same time, the historical zone around the city (managed as such) was seen as a host site for appearance-repair style conservation, as they had yet to receive as many conservation supports as had the national heritage buildings. The sites for demolition were assigned discretely across the village. Residents stated that these sites could be demolished and transformed to improve the harmony between public spaces, infrastructure, and the overall landscape of Huangshan traditional village.

Via the DBSCAN clustering algorithm, we mapped all gathered data points for our micro-regeneration interventions, and clustered these into six spatial groups based on their area densities. These results provided an excellent planning guide for the future development of rural micro-regeneration strategies. In our Huangshan village case, the rural renewal process referred to the opinions of multiple stakeholders, including authority and decision-makers (local governments), professionals (planning consultants), capital holders (tourism companies), visitors, and residents. Among them, the residents who live around the six cluster areas will experience the biggest transformation thanks to community rebuilding, house transformations, and landscape changes that are likely to occur in and around their buildings. It is therefore of critical importance that professionals and decision-makers assess resident perceptions and preferences when creating rural micro-regeneration strategies, and should also note resident relationships and the social character of individuals to better understand the driving factors which affect those perceptions and preferences. The results of our assessment are presented in the next section.

### *4.2. Assessing the Perceptions and Preferences of Residents When Creating a Rural Micro-Regeneration Strategy*

#### 4.2.1. Characteristics of Respondents Who Live in Clustered Areas

Overall, we analyzed 150 residents (54% male and 46% female) who live in the clustered zone in Figure 3B amid the four proposed micro-regeneration interventions. The respondents have mostly been recorded as agricultural people (97.3%) as per the HUKOU (a system of household registration used in China), including five unemployed people and three village committee administrators. A proportion of 19.6% of respondents were under 35 years old, while 53.7% were between 30 and 60; 26.7% of residents were over the age of 66. According to a statement from the National Bureau of Statistics of China [39], young respondents are classified as being under 35, to differentiate from local elderly populations. Most respondents (73%) had lived in Huangshan village for more than 20 years. Respondent income demographics also reveal a diverse income proportion, where the majority (37.3%) see an average annual income of lower than 10,000 RMB per year, closely followed by residents (20.7%) with incomes ranging from 20,000 to 30,000 RMB per year. Very few respondents (4.7%) collect an income in the highest bracket of more than 70,000 RMB. As a result, the average income for this segment of the local population is 40,000 RMB per



year, with 27.3% of residents collecting incomes above that level. Respondents were then classified into either high- or low-income groups.

### 4.2.2. Public Perceptions and Preferences of Village Transformation

Because most (if not all) village transformation outcomes for any future rural micro-regeneration strategy would manage the redevelopment or repair of the very buildings our respondent-residents occupy, any such interventions demand analysis of resident perceptions and preferences as they relate to the restoration activities that will occur around them. We therefore in this study explored (1) a potential mode for optimized management, and (2) the demand for public infrastructure improvements considering future village transformations as per residents' perceptions and preferences for both physical environment improvements and social community management efforts.

#### Potential Management Modes

Based on face-to-face interviews with respondents, we summarized qualitative and quantitative answers into three main management methods in accordance with resource and financial supports: (1) unified repair through government subsidies; (2) tenant repair through house rentals, for example, buildings rented to homestay companies who repair the place to improve business; and (3) self-financed repair. In general, respondents showed diverse preferences and perceptions concerning management methods. Indeed, the majority of residents (46.7%) preferred unified repair through government subsidies for future micro-regeneration interventions. This preference was followed by self-financed repair (36.7%), while respondents expressed less interest in tenant repair through housing rentals (18.7%).

We then tested for significant differences between respondents of different genders, ages, and incomes as per their preferences on micro-regeneration management modes. A Chi-square test indicated that no significant differences between young and old respondents exist for their perceptions of each management model ($\chi^2$ = 2.925, $p$ = 0.142, n = 150), while gender and respondent income did affect participant responses and perceptions.

Of all the respondents, female and male respondents expressed similar interests in unified repair through government subsidies (female 51%, male 43%) and self-financed repair (female 39%, male 31%). On the other hand, males showed more openness (26%) to the tenant repair through rental house categories than their female counterparts (10%). A Chi-square test also indicated significant differences between female and male respondents per rental house for tenant repair as a method of restoration and regeneration ($\chi^2$ = 6.1119, $p$ = 0.013, n = 150) (Figure 4).

Concerning their economic statements, respondents in the high-income bracket showed a greater interest in restoring old buildings via self-financed repairs (51%) and tenant repairs via rental houses (47%). This subgroup expressed strong perceptions and preferences on tenant repairs by rental houses when compared with respondents in lower-income brackets, who have no real interest in this management model. For the majority of respondents on a low income, unified repair through government subsidies is the most popular method for regeneration strategy management (76%).

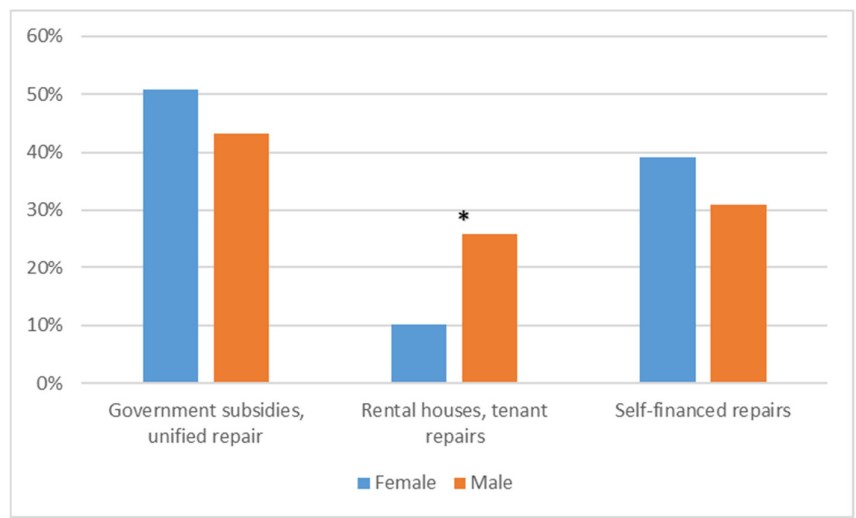

**Figure 4.** Variations in management mode as perceived and preferred by respondents by gender. Asterisks indicate significance level: * means $0.01 \leq p \leq 0.1$.

Finally, a Chi-square indicated that the income of respondents did produce significant differences between their perception and preferences of the unified repair by the government subsidies ($\chi^2 = 79.027$, $p = 0.000$, n = 150) option, the tenant repair through rental houses ($\chi^2 = 53.098$, $p = 0.000$, n = 150) option, and the self-financed repair ($\chi^2 = 11.242$, $p = 0.000$, n = 150) option (Figure 5).

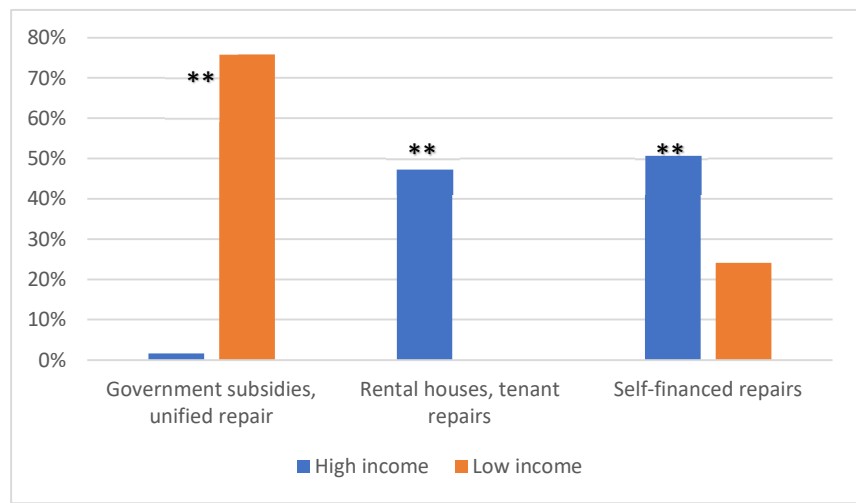

**Figure 5.** Variations in management mode as perceived and preferred by respondents concerning different income levels. Asterisks indicate significance level: ** means $p \leq 0.01$.

The Demands for Public Infrastructure Improvements

Respondents have also expressed their various demands for public infrastructure improvements. Based on a qualitative analysis of participant responses, we summarized such demands for public infrastructure improvements under three main headings: (1) public green spaces, for example, pocket parks for rest, cultural squares for recreational activities, and street greening; (2) cultural service facilities like a library and/or cultural center for villagers; and (3) traffic condition improvements to roads, rural bus stops, and even shared bicycle parking. Of all the public infrastructure demands, 35.5% of respondents preferred public green spaces, while 50% paid more attention to traffic condition improvements. Neither group showed a major interest in the improvement of public cultural service facilities (14.7%).

To explore the significant differences between respondents with different social characters at the intersection of their demands for public infrastructure improvements, we then performed a Chi-square test. Our Chi-square test indicated no significant differences between young and old respondents per their demands for public infrastructure improvements ($\chi^2 = 1.925$, $p = 0.282$, n = 150). The gender and income of respondents did, however, affect their perceptions of the requirements inherent to public infrastructure improvements.

In general, female respondents (41%) preferred public green spaces, while male respondents regarded the improvement of traffic conditions (62%) as their primary infrastructural demand. Female respondents (23%) also expressed a greater demand for improvements to cultural service facilities over males (7%).

A Chi-square test indicated a significant difference between female and male respondent preferences about cultural service facilities ($\chi^2 = 7.414$, $p = 0.006$, n = 150) and traffic conditions ($\chi^2 = 9.689$, $p = 0.002$, n = 150) as well (Figure 6).

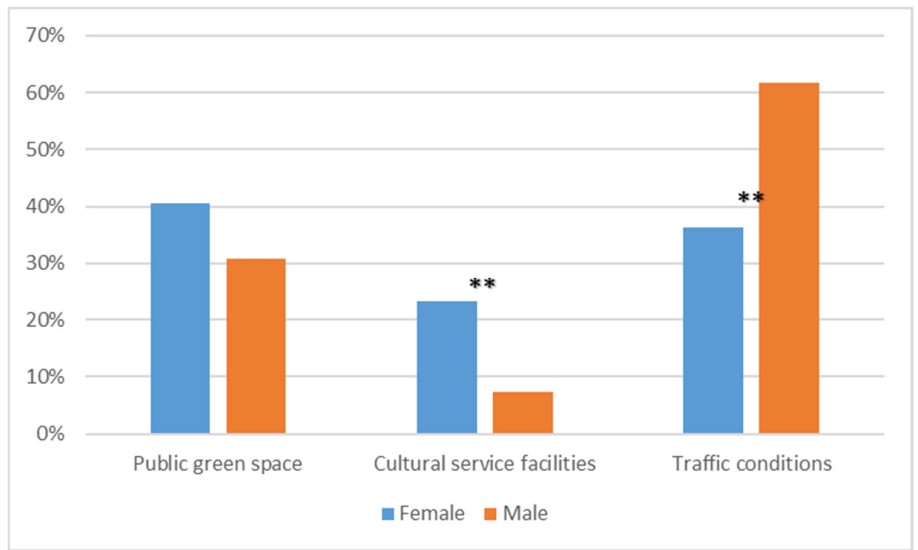

**Figure 6.** Variations in demand for public infrastructure improvements as perceived and preferred by female and male respondents. Asterisks indicate significance level: ** means $p \leq 0.01$.

Moreover, the demands of public green space improvements were mostly perceived and preferred by respondents in the high-income bracket (61%). This majority was followed closely by similar demands for traffic condition improvements (31%), which are the most important public infrastructure demands for respondents in lower-income brackets (63%). A Chi-square test revealed that significant differences exist between residents of differing income levels with respect to all demands for public infrastructure improvement, including: public green space ($\chi^2 = 28.076$, $p = 0.000$, n = 150), cultural service facilities ($\chi^2 = 2.979$, $p = 0.084$, n = 150), and traffic conditions ($\chi^2 = 14.777$, $p = 0.000$, n = 150) (Figure 7).

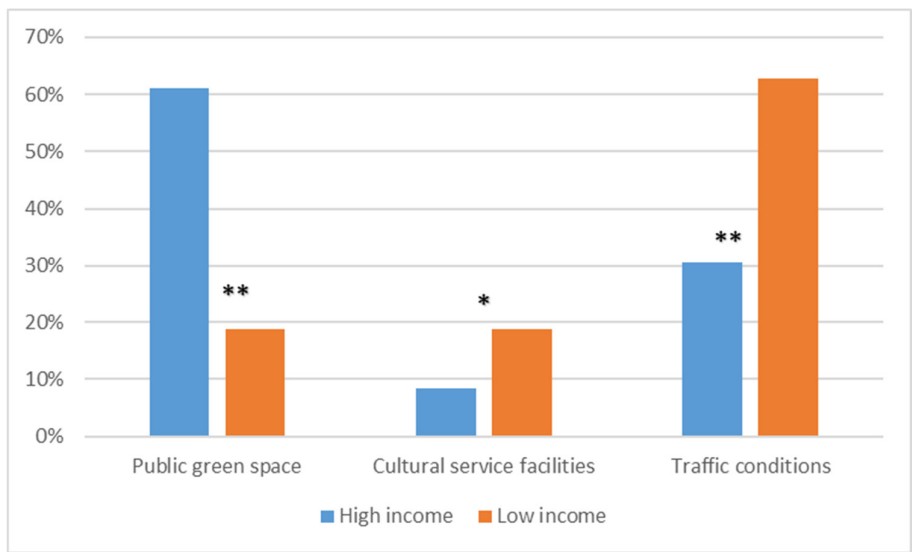

**Figure 7.** Variations in demand for public infrastructure improvements as perceived and preferred by respondents of varying income. Asterisks indicate significance level: ** means $p \leq 0.01$, * means $0.01 \leq p \leq 0.1$.

### 4.2.3. Public Perceptions and Preferences of Local Industrial Development

Considerations about the importance of local industry during the development phase of any rural micro-regeneration strategy, and assessments of resident perceptions and preferences as they relate to four main avenues of industrial development, are critical for sustainable development in the future.

Generally, the majority of residents (52%) prefer tourism over other industrial developments. For more traditional industries like cultivation, there are still 20.7% of residents who show a preference here concerning their village's industrial future. By contrast, E-commerce is preferred by 14% of residents as the village's primary economic future, while even fewer residents (13%) showed a preference for hand-workshops. Several factors significantly affected each respondent's perception of and preferences for the industrial future of their village. Based on the Chi-square test, we indicated that gender, age, and average annual income played a significant role in how respondents ranked their preference for the four industries.

Firstly, the gender of respondents made a significant difference and accounts for how residents perceived and preferred the future of that industry (Figure 8). Female respondents preferred tourism (63.8%) closely followed by a preference for cultivation (18.8%). Male respondents also perceived tourism (52%) as the most popular industry for development, and showed a higher interest in E-commerce (25.9%) and cultivation (22.2%) interventions than their female counterparts (0%, 18.8%). Our Chi-square test revealed a statistically significant difference between female and male respondents with respect to their perception and preferences on tourism ($\chi^2 = 7.089$, $p = 0.008$, n = 150) and E-commerce ($\chi^2 = 7.089$, $p = 0.000$, n = 150) developments.

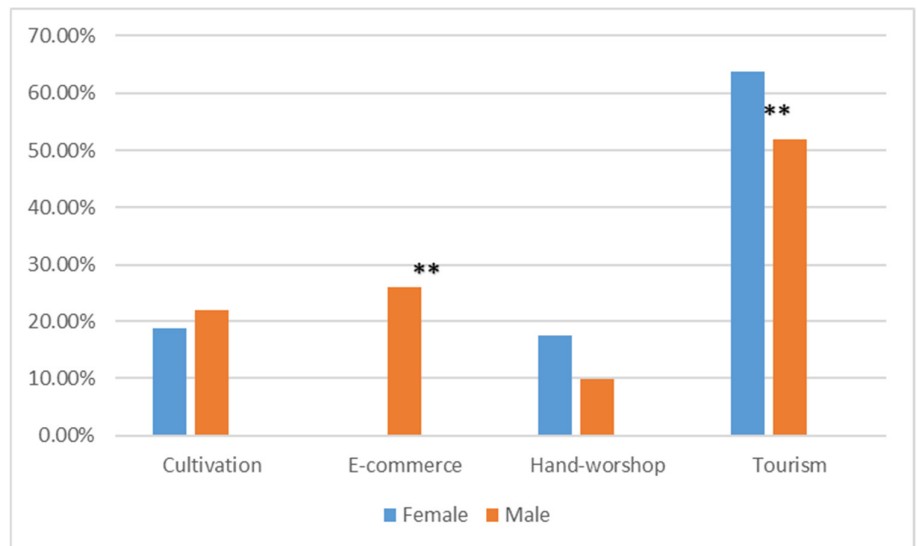

**Figure 8.** Variations in future industry, as perceived and preferred by female and male respondents. Asterisks indicate significance level: ** means $p \leq 0.01$.

The age of respondents also affected respondent's perceptions and preferences on the four types of industrial development. Even though tourism is widely regarded as the industry which most demands future development by both young (76%) and senior respondents (46%), there are still significant differences that exist between young (residents under 35) and old residents (over 35). Younger residents show no interest in traditional industrial developments such as cultivation and hand workshops; on the contrary, they expressed a much higher preference for E-commerce development (23%) than did their older counterparts (12%). Here, the Chi-square test revealed a statistically significant difference between respondents of different ages with respect to their perceptions and preferences on cultivation ($\chi^2$ = 8.193, $p$ = 0.004, n = 150), hand-workshops ($\chi^2$ = 4.838, $p$ = 0.028, n = 150), and tourism ($\chi^2$ = 7.827, $p$ = 0.005, n = 150), as found in Figure 9.

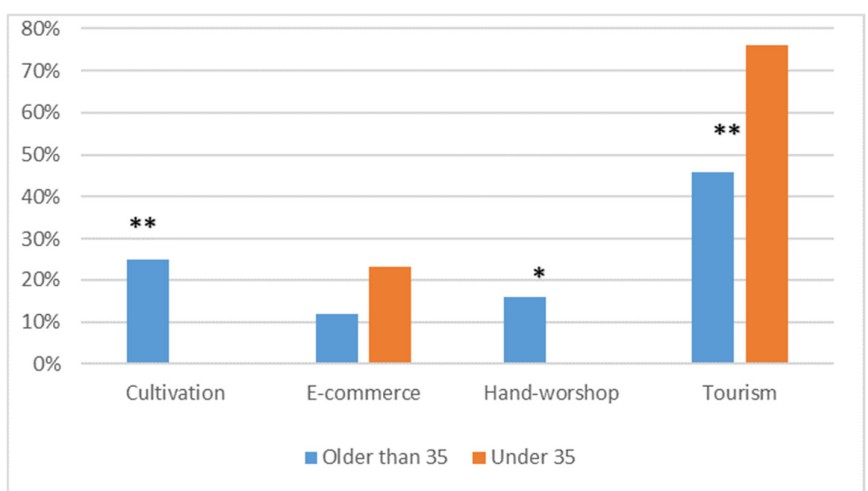

**Figure 9.** Variations in the future industry perceived and preferred by young (under 35) and old respondents (over 35). Asterisks indicate significance level: **means $p \leq 0.01$, * means $0.01 \leq p \leq 0.1$.

Moreover, concerning the future industry, the income of residents significantly affects their perceptions and preferences as well (Figure 10). Indeed, when we classified the respondents into low- and high-income groups based on local average annual income (40,000 yuan per year), respondents in the low-income bracket expressed a more pronounced preference

for tourism (41%) and cultivation (34%) improvements, while respondents with a high income expressed no interest in cultivation. Instead, these populations place their focus on tourism (69%) and E-commerce (29%) developments, especially when compared to their low-income peers. The Chi-square test also revealed that the income of respondents highly affects their perception of and preferences for each of these four industries. There are statistically significant differences between respondents of differing incomes with respect to their preferences for cultivation ($\chi^2 = 25.3340$, $p = 0.000$, n = 150), E-commerce ($\chi^2 = 17.725$, $p = 0.000$, n = 150), hand-workshops ($\chi^2 = 11.399$, $p = 0.001$, n = 150), and tourism ($\chi^2 = 11.92$, $p = 0.001$, n = 150).

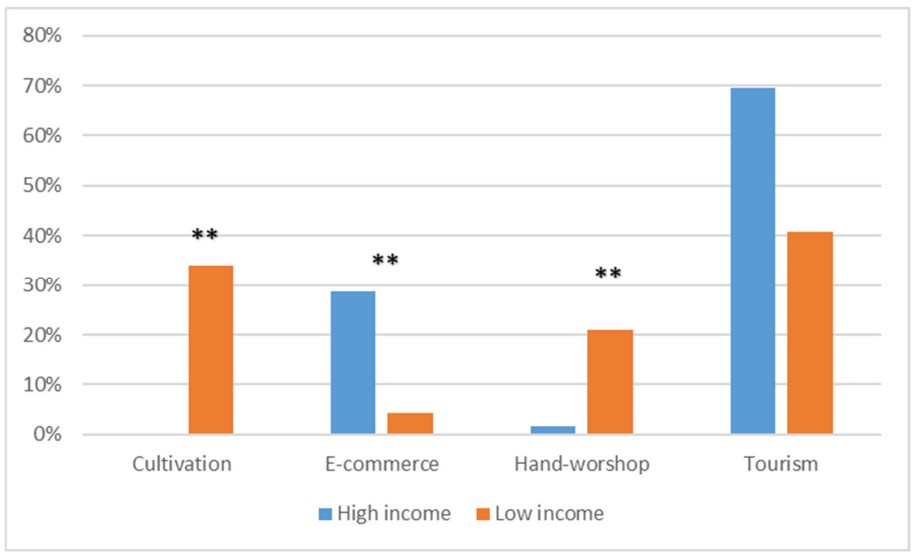

**Figure 10.** Variations in future industry, as perceived and preferred by respondents from high- and low-income groups. Asterisks indicate significance level: ** means $p \leq 0.01$.

## 5. Discussions

This study set out to explore the vast potential associated with the promotion of micro-regeneration strategies based on assessments of public perception and preferences, specifically in the context of an "unlisted" traditional village in China.

### 5.1. Academic Implications for Traditional Village Conservation and Development

Within the context of a national rural renewal policy, traditional village conservation studies have attracted increased attention to the protection of local cultural landscapes and enhanced social wellbeing [40–42]. However, formal conservation governance in traditional villages in China has often been focused on an overall master plan which regards an entire village's physical style. Such plans often control many aspects of industrial layout planning [43–45], which are a required investment for the improvement of a village and its cultural landscape.

For most "unlisted" traditional villages, there are limited investments and resources available to support their unique conservation and development needs or suggested projects. Thus, rural micro-regeneration strategies, which focus on small-scale interventions, have become a potential means by which even a small, lightly funded rural traditional village may govern their own renewal.

Our first results indicated a spatial cluster of sites that demanded small-scale interventions from local public perspectives on micro-regeneration strategies. From demolition to repair, the adoption of flexible small-scale spatial interventions to formulate targeted protection strategies were based on public perspectives for parts of the village, and were compared with formal traditional village conservation practices in China. Residents are keenly aware of the historical value and heritage of their village, and have close knowledge

of which sites must be kept or restored. However, no rural micro-regeneration strategy that had been based on resident perceptions had been studied in the past; indeed, literature about public participation in rural regeneration was mainly focused on policy or regional development. The small amount of previous research similarly stresses either sustainability or community participation.

In the context of resident perspectives, this study is among the first to develop an integrated cultural perspective approach in the context of traditional village revitalization. Our study does intersect with current literature in its emphasis on micro-regeneration strategies that rely on resident preferences and perceptions, and which note that these as key to village revitalization and sustainable development. This study contributes to that literature and extends our knowledge capital regarding an integrated economic regional renewal approach to the revitalization of traditional village success and rural landscape conservation, as well as to the improvement of local industrial development. For this reason, among others, Huangshan's future regarding possible micro-regeneration strategies sets a practicable example for the overall success of rural renewal and sustainable development initiatives across the globe.

Against this backdrop, our interest was in how residents who live around the sites where future interventions are planned view the future of industrial development and village transformation via micro-regeneration strategies, which are intended to underpin traditional village conservation and industrial development.

### 5.2. Public Perceptions and Preferences on Village Transformation for Rural Micro-Regeneration

In terms of village transformation, the most popular management model perceived by respondents as being a top preference was unified repair through government subsidies, while the weakest interest was in tenant repair through housing rentals. When local respondents were asked about their preferences on village transformation, we found that they more often had high expectations on government intervention and guidance and expressed ignorance about rural housing rental methodologies. Additionally, our results revealed that traffic conditions were perceived as a high-priority demand for public infrastructure improvements, followed closely by public green spaces. Public cultural service facilities were rarely preferred by local respondents. Our study also explored whether gender, age, and income level affect respondents' perceptions and preferences on the management of public infrastructure improvements that support village transformation. We emphasized that there are no significant differences made on the axis of age, while the gender and income of respondents do affect how they perceive village transformation.

Existing literature also addressed the effects of gender on a resident's willingness to engage in local landscape conservation and sustainable development per documented rural studies [46–48]. Previous research has indicated that men and women often have different relationships with their surrounding environments due to the gendered divisions of labor, sense of place, resources, and responsibilities [47,49,50]. However, the dimension of gender has often remained unconsidered in the literature concerning micro-regeneration and land-use development. Our study further indicated that gender variables made significant differences per resident preferences and perceptions on traditional village transformations that rely on rural micro-regeneration strategies. For instance, while women expressed less interest in tenant repair through rental housing, they showed a stronger preference for public green spaces and cultural service facilities for public infrastructure improvement. Women respondents also stated that micro-enhancements to public green spaces would provide the greatest benefits to recreation, ecology and aesthetic value, and would sustain positive health impacts. Further still, their demands for cultural service facilities are primarily directed toward social activities and education, for example, rural children's libraries. These results agree with previous studies that relate female perceptions as commonly attributing stronger importance to aesthetic value, social relations, and environmentally sustainable infrastructure management [48,51]. A consideration of gender may thus target a more comprehensive scope of human need in the context of micro-regeneration, all in the

pursuit of space-quality improvements. In sum, our findings would enhance a national and global understanding of the complexity inherent to the relationship between individuals and community, which may then lead to better conservation practices and policies as they relate to micro-regeneration strategies surrounding traditional village conservation.

Our results also highlight that the income of respondents is an important influencing variable for how resident perceptions impact preferences and demands for public infrastructure improvements and management models. Our study indicated that respondents with higher incomes had a significantly stronger appreciation of public green space as a public infrastructural demand. Similar results have been observed for other environmental issues and projects, such as for water conservation and environmental protection [52,53]. Previous studies have stressed that residents with a higher income will have stronger perceptions regarding the environment. [54–56]. However, the impacts of income have not explicitly been explored in any micro-regeneration studies known to date. Our findings therefore enhance a potential understanding of the impacts that public economic status has on rural micro-regeneration, and will benefit traditional village conservation practices in the future.

Moreover, our study indicated that residents with a high income also pay closer attention to self-financed and tenant repair strategies (the latter via rental houses) as management models when compared to government repair strategies that focus on a unified village style. Many scholars have addressed the trend that individuals with higher incomes often show stronger self-awareness of, and participation in, public affairs practices and policies [57]. In our study, we therefore extended our public perception and preference measures to include the similar influence of income relating to the planning of rural transformation and micro-regeneration strategies. During these interviews, respondents stated that even though they desire harmony in the appearance of local buildings in the traditional village, self-financed repair methods would directly turn their ideas and opinions for living spaces into action, while allowing them to be flexible to their personalized residential needs. This is especially true when you compare this method to a more unified repair model designed by the government. Others preferred the option of renting their buildings to professional homestay organizations, who could then provide professional homestay management experience and a unique and good restoration protocol for aging buildings. For example, a widely known homestay brand called NAKEHOME rents local, aging buildings in villages across Zhengjiang Province, and produces replicable, unique architectural design and restoration approach for each building they own. Respondents in this study stated that the potential benefits to recreation, aesthetic value, and the positive health impacts of green space access are why they regarded these priorities as such. On the opposite end of the spectrum, lower-income residents in our study preferred government repair strategies that rely on old building reconstruction and subsidized small-space development. Still, some literature has indicated that low-income people always tend to have their own preferences when it comes to living spaces, which leads to a disconnect between what the government wants and the demands of the low-income people who live in a given area [58]. Apart from that, in our study low-income residents presented with a strong willingness to support government measures that will improve local living spaces. They supported a unified style design as well, which ultimately extends the existing knowledge base on how low-income persons perceive their residential or community environment. In closing, this study could be regarded as the first to overtly indicate how resident income can affect the perceptions and preferences of those residents in terms of micro-regeneration village transformation strategies. Our findings are expected to be a major consideration for those who intend to design a micro-regeneration strategy that will target residents of varying income classes.

*5.3. Public Perceptions and Preferences on Rural Micro-Regeneration Industrial Improvements*

The highest-ranked future industry for collected respondent perceptions and preferences is tourism. Similar findings have been confirmed by the booming rural tourism

industry, which actively promotes rural development [59–62]. Respondent data also emphasized the foundations of tourism development, including (1) rich cultural resources and heritage sites such as the 600-year-old 'Eightside' Hall as a part of national heritage; (2) beautiful natural environments, such as the traditional south Yangtze region cultural landscape; and (3) locations that are a short distance from big cities. Participants further stated the importance of such benefits as enhancements to income and the improvement of their village's scenic quality. Hand-workshops were, however, rated amongst the least preferred improvement option, with participants stating, "our current hand-made workshops are scattered, and lacking in large-scale production". For cultivation, the increased agricultural costs and risks associated with a changing climate and market, low land output, and limited arable land per capita similarly reduced interests in cultivation, especially traditional paddy fields. The remaining reasons respondents prefer cultivation include agricultural subsidies and cultivation plans for cash crops. Interestingly, cultivation was considered more than even E-commerce, a result that seems to stray from the studies that attest to increasing rural E-commerce developments in China [63,64].

Respondents stated that the "complex" operation and transaction procedures and "strict" requirements for products on the current E-business platforms are great obstacles to rural economic E-commerce success. "Social media like Kuaishou and TikTok are very little help as well, we have little followers," they stated. Furthermore, current E-commerce transportation systems have not proven convenient, which requires more E-commerce education on both a technological and product management scale, to enhance respondents' perceptions of E-commerce as a viable micro-regeneration strategy. Besides, logistical traffic infrastructural development is already in demand as a local industrial avenue for development.

Our study thus indicates that gender, age, and income level significantly affect and play a role in respondents' perceptions about and preferences for future industrial developments in their area. Our results showed that males had a stronger appreciation for E-commerce futures than females. Furthermore, during our face-to-face interviews, we found that the current E-commerce transportation system requires residents to carry agricultural products to the town for trading, which reduces access opportunities for female market participants. In contrast, females have expressed, in this study, a higher preference for tourism industrial development, a factor which is consistent with previous studies that have linked female empowerment with rural tourism development [65–67].

In this context, the development of a village's tourism industry could become an effective way to empower local women via increased income and participation opportunities. Besides, our findings have identified significant age differences in industry perceptions and preferences. We found that older people attributed a higher preference to cultivation and hand workshops, while younger respondents showed little interest in these. The loss of interest in cultivation and traditional hand workshops among young people is often ascribed to urbanization, industrialization, and westernized urban cultures [68–70], which indicates that efforts are needed to strengthen the education and transmission of traditional knowledge and cultural confidence in conservation strategies. Moreover, we demonstrated that income level does make a significant difference in the context of industrial development from a public perspective. Respondents with higher incomes showed a stronger interest in E-commerce over those who set their focus on a more traditional form of agricultural industry improvement; in fact, many stated that, despite the current transportation system barriers for rural E-commerce, they have a high-value perception of the potential of E-commerce that is based on the information they had received to that point. Any further information gaps may thus affect those of different income brackets, causing varied perceptions of E-commerce, which in turn creates significant barriers to development. This outlook differs from Lawrence's [71] statement that the absence of basic infrastructure and the lack of government guides and national ICT strategies and are significant barriers for low-income people in developing countries.

## 6. Conclusions

Micro-regeneration strategies promote environmental improvements to the physical and social aspects of space through small-scale interventions. Our study therefore provides a sight into how we may consider rural micro-regeneration strategies as a potential to accommodate "unlisted" Chinese traditional villages via the emerging rural renewal opportunities.

The study outlines the importance of local public perceptions and preferences as an important complement to current micro-regeneration literature, which is often aimed at enhancing residents' well-being in both physical and social environments, but without much knowledge of resident interests or population outreach. We regard this research as an initial step toward an improved understanding of resident preferences when it comes to the conservation and revitalization of traditional villages in China, and across the globe. It also offers inspiration for those who intend to practice real-world rural micro-regeneration strategies and renewal policies common to the Chinese rural cultural context.

A final important insight from our study is that the unified repair of buildings and spaces through government subsidies, improvements to traffic conditions, and developments in the tourism industry are the most approved directions across respondents for local rural micro-regeneration strategies. Our survey also revealed that age, gender, and income significantly affected such interests; while the former affects the perceptions and preferences of industry, the latter affects how people perceive both the industrial and transformational aspects of such micro-regeneration strategies. Women showed significantly strong favoritism for tourism and public green space improvements, which is also a trend among residents with high incomes. Residents with a low income are on the other hand the main supporters of unified repair through government subsidies, and often place their preferences on traffic condition improvements. Overall, the key holders of traditional cultivation (e.g., paddy field) and hand workshops (e.g., root carving, yellow rice wine production) are in the older age group, while younger residents were more cognizant of emerging industries.

We thus end our study with a reflection on its implications for rural renewal and traditional village conservation practices. We conclude that rural micro-regeneration strategies based on resident perceptions and preferences provide an accessible and economic path by which these plans may contribute to traditional village conservation and development in the context of rural renewal. Cooperative planning with residents deserves greater awareness and consideration in rural micro-regeneration strategies, and a deeper and more detailed analysis is needed to properly assess the performative success of such micro-regeneration strategies—especially those that address traditional village conservation. A fuller understanding is also required regarding the empowerment of female respondents through rural tourism, the education of low-income villagers for integration into E-commerce sectors, and the attraction of young people to regional traditional skills, ecological knowledge, and cultural landscape conservation to stimulate local vitality. These issues also demand a mobilization of resources, social power, and capital (as well as the creation of a broader set of interests) from governments, NGOs, organizations, experts, and tourists. Altogether, the type of resident engagement practiced in our study provides an improvement to the overall sustainability and acceptance of micro-regeneration strategies and rural economic development in China, and we look forward to studying the implications of this fact during future research.

**Author Contributions:** Conceptualization, X.X. and H.X.; methodology, H.X.; software, G.Z.; validation, G.Z., and X.X.; formal analysis, G.Z. and Q.Z.; investigation, Q.Z.; resources, X.X.; data curation, Q.Z.; writing—original draft preparation, H.X.; writing—review and editing, H.X.; visualization, G.Z.; supervision, H.X.; project administration, X.X.; funding acquisition, X.X. All authors have read and agreed to the published version of the manuscript.

**Funding:** This research received no external funding.

**Informed Consent Statement:** Informed consent was obtained from all subjects involved in the study.

**Data Availability Statement:** Not applicable.

**Conflicts of Interest:** The authors declare no conflict of interest.

**Appendix A**

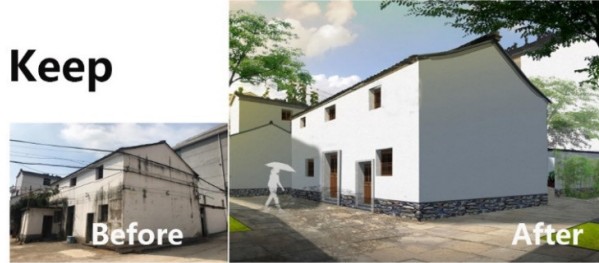

Keep the original style.
Maintain with cultural relics protection law.
Improve surrounding environment

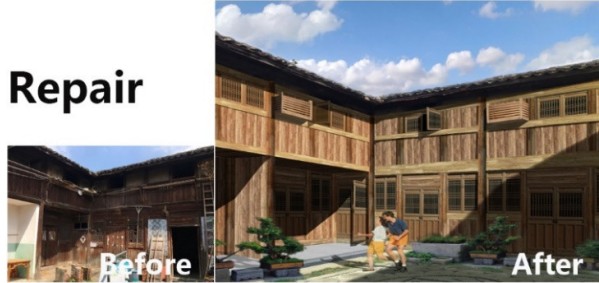

Repair appearance and strengthen
the buildings through raw materials and
traditional techniques.

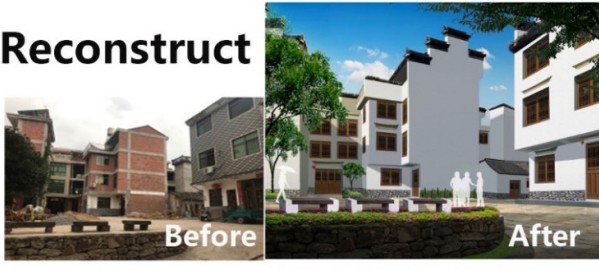

Reconstruct the function and basic facilities.
Enovate the building facade to make it unified
with the traditional village

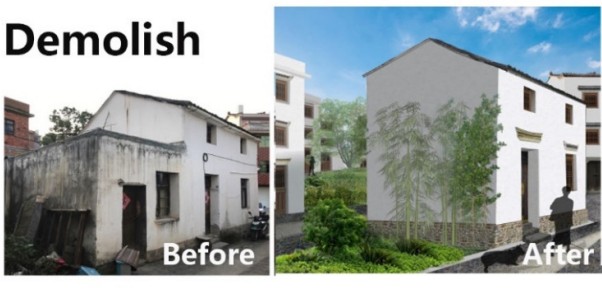

Demolish the buildings and transform them
into pocket green space for public.

**Figure A1.** Scenarios to describe four micro-regeneration intervention types.

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
