# Peer review of "Making Rural Micro-Regeneration Strategies Based on Resident Perceptions and Preferences for Traditional Village Conservation and Development: The Case of Huangshan Village, China"

_land, doi:10.3390/land10070718_

Round 1

Reviewer 1 Report

The manuscript presents an interesting research which concerns an important and somewhat understudied topic - the conservation and revitalization of the “unlisted” traditional villages with a Chinese context, one appropriate because the country’s rich cultural heritage endowments and huge stock of such villages. The notion of micro-generation strategies is also interesting, and in Section 2 the authors have made convincing arguments on the importance and applicability of the strategies in the field of traditional village revitalization. In the sections that follow the authors have presented a case study. Overall, this is a fair piece. However, I still have the following question, which I would suggest the authors respond before the manuscript is forward to publication process.   1. In introduction, the authors stressed two major caveats when it comes to the revitalization of traditional villages, one regarding the perceptions and preferences of villagers, the other regarding an activation protocol as the stimulator for industrial and economic development and also community relations. I suppose that the main intention of the research is to respond to existing research gaps concerning the two points. The authors did touch upon these points throughout the manuscript, however, I failed to find specific, clearly summarized research gaps, as well as research questions and hypotheses that address them. I recommend the authors specifically state the research gaps, research questions that address them, and respective hypotheses.   2. The authors mentioned stakeholders in framing the research, which is good. However, in the empirical part, that who are the stakeholders in this case is not clearly presented. For example, are the two genders or people of different economic status considered different stakeholders, or are they governments of various levels, capital holders, etc.? Please clarify.   3. Further, why are the group difference tests performed with respect to the residents gender and economic status? Are there literature or field survey findings showing that they are the key variables that divide people’s preferences and perceptions on strategies of rural revitalization in a process of micro-regeneration? Additional literature review and discussion may be needed to make the hidden assumptions clear. Also, the authors may want to discuss the implications of the empirical findings regarding the aforementioned issue in the Discussion Section, such that the methodological and intellectual contribution of the work be made clearer.   4. On the “activation protocols”, are they the four ways of “management” in the empirical part? Are these four ways purely hypothetical ones with no further details, or is there a real planning that designates particular means of development? Either way, how have the authors ensure that the interview participants consistently and correctly understand the means of the particular ways?   5. In Fig. 1, what are the numbers in the legend for “clusters” mean? It appears to me that they are merely dot density values, in which case only up to three significant figures are enough. Also, a mere dot density analysis would not automatically present the clusters. Supplemental methods such as the DBSCAN clustering algorithm may be considered.   6. There are minor language/style errors in the manuscript. For example, Figs. 3 and 4 are regarding the same dependent variable, but are with different caption titles; In Fig. 5, the unformatted “Axis Title” notation should be removed; some paragraphs, such as in lines 303, 385, 469-471, etc. do not look correct. Please address these problems.

Author Response

Thank you so much for the feedback and the suggestions for our manuscript. We apologize for the incomplete form of the previously submitted manuscript. We have fully addressed your suggestions, and have sought to create a concise, consistent, and clear manuscript. Please see our point-by-point responses to the reviewers, below.

Reviewer 2 Report

First, I would like to congratulate the authors for the paper, which is very interesting and whose theme is totally related to the Land journal. The investigation is clear, well planned and well resolved, as well as the methods have been intelligently chosen and applied. I love the method used to capture and represent, on the territory, the opinion of the interviewed actors. Moreover, I consider that the structure of the paper is correct, as well as the abstract and the discussion section.

However, I would like to make some recommendations that could increase the quality of the paper presented. First, authors should more clearly state the research hypothesis and, secondly, Kernel Density method should be better and more deeply explained relying on bibliographic references.

Best wishes.

Author Response

(The authors gave the same response as above.)
